# Organophotocatalytic dearomatization of indoles, pyrroles and benzo(thio)furans via a Giese-type transformation

Yueteng Zhang[1], Peng Ji[1], Feng Gao[1], Yue Dong[1], He Huang[2], Changqing Wang[1], Ziyuan Zhou[3] & Wei Wang [1✉]

Accessing fascinating organic and biological significant indolines via dearomatization of indoles represents one of the most efficient approaches. However, it has been difficult for the dearomatization of the electron deficient indoles. Here we report the studies leading to developing a photoredox mediated Giese-type transformation strategy for the dearomatization of the indoles. The reaction has been implemented for chemoselectively breaking indolyl C=C bonds embedded in the aromatic system. The synthetic power of this strategy has been demonstrated by using structurally diverse indoles bearing common electron-withdrawing groups including (thio)ester, amide, ketone, nitrile and even aromatics at either $C_2$ or $C_3$ positions and ubiquitous carboxylic acids as radical coupling partner with high *trans*-stereoselectivity (>20:1 dr). This manifold can also be applied to other aromatic heterocycles including pyrroles, benzofurans and benzothiophenes. Furthermore, enantioselective dearomatization of indoles has been achieved by a chiral camphorsultam auxiliary with high diastereoselectivity.

[1] Departments of Pharmacology and Toxicology and Chemistry and Biochemistry, and BIO5 Institute, University of Arizona, Tucson, AZ, USA. [2] Department of Chemistry and Chemical Biology, Cornell University, Ithaca, NY, USA. [3] National Clinical Research Centre for Infectious Diseases, Shenzhen Third People's Hospital, The Second Hospital Affiliated to Southern University of Science and Technology, Shenzhen, China. ✉email: wwang@pharmacy.arizona.edu

The indolines have fascinated organic and medicinal chemists for decades[1–8]. The molecular architecture is a common core featured in numerous natural products, biologically active compounds particularly pharmaceutics and agrochemicals. This biogenically produced privileged structure[9] provides highly biologically relevant three-dimensional chemical space for effective interaction with biological targets. Therefore, quickly accessing the framework with the capacity of engineering functional and stereochemical diversity can streamline the target- and diversity-oriented synthesis for biological studies and drug discovery.

The dearomatization of arenes has become a powerful platform for the facile construction of highly valued molecular architectures[10–15]. The dearomatization of indoles constitutes the most efficient strategy for accessing indolines[1–8]. Indole is an electron-rich aromatic system containing enamine embedded $C_2$–$C_3$ π bond and strong nucleophilic $C_3$ carbon. The reactivity has dictated indole dearomatization methodology development[1–8] since Woodward's pioneering study using a Pictet-Spengler type reaction to break the aromatic tryptamine in total synthesis of strychnine in 1954[16]. Impressively, this important array of reactivity from the intrinsically nucleophilic indoles upon activation by various tailored electrophiles has become a powerful manifold for the synthesis of structurally diverse indolines as it enables regioselective reactivity, facile ring formation, and efficient skeleton rearrangement[1–8]. Moreover, this reactivity has been leveraged beyond the 2e transfer pathway. Single-electron transfer (SET) involved oxidation-induced C–H functionalization of the nucleophilic indoles has been elegantly realized as powerful alternatives for indole dearomatization[17], particularly mild, green visible light photocatalytic and electrochemical methods (Fig. 1a)[3,15,18–29].

Despite the great success, it has been challenging for the dearomatization of electron-poor indoles, as evidenced by only a handful of examples[30–33], which rely on ionic activation mode. Indoles bearing electron-withdrawing groups (EWGs) at N, $C_2$ or $C_3$ positions tend to make the $C_2$=$C_3$ π bond more difficult to react with electrophilic partners or in an oxidative SET process. The reduced $C_2$=$C_3$ π bond electron density can be reflected by the significant difference of their redox potentials. For example, $E_{redox}$ of N,3-dimethyl indole is ca. +0.4 V vs SCE[34], while N-methyl 3-acetyl indole is ca. +1.0 V vs SCE[34]. Therefore, an unique activation paradigm is needed to address this unmet synthetic challenge.

Giese reaction involving the reductive conjugation addition of radicals to electron-deficient C=C double bonds servers as a powerful tool for new C–C bond formation[35]. The original conditions using stoichiometric amounts of trialkyl tin reagents have motivated organic chemists to develop more practical protocols. Recent efforts on the study of photoredox catalysis under mild reaction conditions have made the process greener and more atom economical (Fig. 1b)[36–47]. In this process, the unsaturated C=C bond is transformed into a saturated C–C bond in a conjugate addition manner. We questioned whether the reaction could be applied for breaking unsaturated C=C bonds embedded in the indole aromatic structure. Specifically, we envisioned the

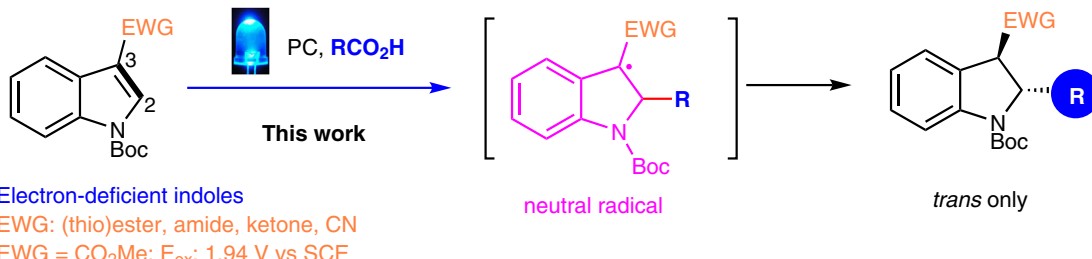

**a. Dearomatization of indoles via oxidative SET of C2=C3 bond (known).**

**b. Giese reaction: radical addition to simple α, β-unsaturated systems.**

**c. Dearomatization of indoles via a Giese-type transformation (this work).**

**Fig. 1 The synthesis of indolines by radical engaged dearomatization of indoles. a** Dearomatization of indoles via oxidative SET of C2=C3 bond (known). **b** Giese reaction: radical addition to simple α, β-unsaturated systems. **c** Dearomatization of indoles via a Giese-type transformation (this work).

incorporation of an EWG into $C_2$ or $C_3$ position of indoles, which could be viewed as the Michael acceptors for the Giese type transformation. The successful realization of this process could offer a distinct approach for the dearomatization of less developed electron-deficient indoles and would also expand the scope of the Giese reaction.

However, implementing the strategy faces significant roadblocks. Unlike an isolated C=C bond in a typical Giese reaction (Fig. 1b), breaking the unconventional C=C bond in stable indole aromatic systems overcomes a higher energy barrier. The precedent studies of direct addition of an electrophilic radical to the electron-rich $C_2=C_3$ bond of indoles in electrophilic aromatic substitution processes provide encouraging possibility[48,49]. Nonetheless, the reversed reactivity of the addition of a nucleophilic radical to an electron-poor $C_2=C_3$ bond of indoles is unknown. Moreover, even though incorporation of EWGs into the $C_2$ or $C_3$ positions of indoles could reverse the polarity from the innate nucleophilic to electrophilic system and serve as a potential radical acceptor, the weakly electron-deficient indoles render the Giese reaction more difficult because more electron deficient, less hindered α, β-unsaturated systems are generally used for effective nucleophilic radical addition[36–47]. Furthermore, in the photoredox process, possible oxidation of the weakly electron-deficient indole systems could complicate the process.

Herein we wish to disclose the results of the investigation, which leads to a photoorganocatalytic strategy for the dearomatization of electron-deficient indoles. An unconventional Giese-type transformation is successfully implemented for the first time (Fig. 1c). Notably, the protocol uses naturally abundant carboxylic acids as radical precursors[50–55] for reacting with various functionalized indoles bearing common EWGs at either $C_2$ or $C_3$ positions including (thio)ester, amide, ketone, nitrile, and even neural H and phenyl moieties. Furthermore, this mild dearomatization method displays a broad substrate scope and a wide array of functional group tolerance and thus enables to deliver a wide array of 2,3-disubstituted indolines with high *trans*-stereoselectivity (>20:1 dr). This approach can also be applied to other aromatic heterocycles such as pyrroles, benzofurans, and benzothiophenes for the dearomatization. Finally, enantioselective dearomatization of indoles has been achieved by a chiral camphorsultam auxiliary with high ee (up to 98%).

## Results

**Exploration and optimization of the Giese-type reaction.** In the initial exploratory studies, we chose *N*-Boc indole methyl ester **1a** as a radical acceptor and Boc-alanine **2a** as a radical precursor for the proposed Giese-type reaction. It is believed that the carboxylate **2a** can be selectively oxidized by the photocatalyst (PC, Ir[dF(CF₃)ppy]₂(dtbpy))PF₆) to give the corresponding radical while the oxidation of the $C_2=C_3$ π bond is difficult because the $E_{1/2}^*$ is +1.21 V (vs SCE)[50] of the PC and **2a** salt $E_{ox}$ is around +1.00 V (vs SCE)[56] while the $E_{ox}$ of the indole methyl ester **1a** is +1.94 V (vs SCE, see Supplementary Methods Section 1.8). Irradiation of a mixture of **1a** (0.2 mmol) with *N*-Boc-Alaine **2a** (0.26 mmol) in the presence of (Ir[dF(CF₃)ppy]₂(dtbpy))PF₆ (5 mol%) and $Cs_2CO_3$ (0.2 mmol) in DMF (0.1 M) under $N_2$ atmosphere with a 5 W blue LED strip was performed accordingly (Table 1, entry 1). Indeed, the desired indoline product **3a** was obtained with yield of 56% after 36 h irradiation without the observed oxidation of **1a** (entry 1). It is also noted that the reaction proceeded highly *trans* selectively. Encouraged by the results, we devoted efforts to optimize reaction conditions. When 4CzIPN was used as a PC[57], a nearly quantitative yield was obtained (entry 2). Probing other parameters such as switching solvent to MeCN (entry 3), shortening time (entry 4) and

lowering the amount of base (entry 5) revealed that the reaction performed in DMF for 36 h with 1 equiv. of $Cs_2CO_3$ and 5 mol% catalyst (entry 2) could give the best reaction yield. As expected, PC (entry 7) and visible light (entry 8) were indispensable for this process. These findings led to establishing the optimal protocol used for probing the scope of an organophotocatalytic dearomatization of indoles.

**Scope of indoles and other heteroaromatics.** With optimized reaction conditions in hand, we first evaluated the radical-engaged dearomatization reactions utilizing various electron-deficient indoles as substrates. As shown in Fig. 2, this methodology serves as a mild and efficient approach for the synthesis of a wide range of 2,3-disubstituted indoline derivatives in high yields (up to 99%) (Fig. 2) (Supplementary Methods Section 1.5 and Supplementary Data 1). Notably, the protocol works for indoles bearing various EWGs beyond ester. Ketone (**3b** and **3c**), amide (**3c** and **3g**), thioester (**3d**) and cyanide (**3f**) can be served to afford broadly functionalized indolines in high yields. Furthermore, commonly used nitrogen protecting groups such as Tos (**3 h**), Bz (**3i**), Ac (**3j**) and Cbz (**3k**) are tolerated very well. Incorporation of various substituents (e.g., MeO, F, Cl, Br) into the benzene ring in the indole skeleton does not affect dearomatization efficiency (**3l**, **3m**, **3n**, **3o**, **3q** and **3r**). Unexpectedly, in addition to electron-deficient indoles, the protocol works smoothly for indoles containing electron neutral H (**3o**) and phenyl (**3p**) moieties. Moreover, instead of EWG at $C_3$, indole possessing $C_2$ ester group also works well (**3q** and **3r**). Remarkably, other aromatic structures such as benzofuran, benzothiophene and pyrrole can attend the process and produce 2,3-dihydro-1*H*-pyrrole (**3s**), 2,3-dihydrobenzofuran (**3t**), and 2,3-dihydrobenzo[*b*]thiophene (**3u**), respectively in high yields. The obtained *trans* products were confirmed by the single X-ray analysis of *trans*-**3k** (CCDC-1994584, Supplementary Methods Section 1.10: Supplementary Tables 1–3 and Supplementary Data 3: CIF file). The unsuccessful indole substrates listed in Fig. 2 suggest that two EWGs are necessary for this process.

**Scope of carboxylic acids.** Next, we probed the structural alternation of carboxylic acids under the optimal reaction conditions (Fig. 3) (Supplementary Methods Section 1.6 and Supplementary Data 1). Again, this strategy provides a preparative power for the synthesis of various *trans*-selective 2,3-disubstituted indolines on account of their easy availability. Naturally abundant amino acids without requiring protection of side-chain functionalities serve as a good source to install α-amino alkyl groups at $C_2$ position of indoline (**3v-3ae**). The mild reaction enables incorporation of highly strained structures (**3ae** and **3ao**). In addition, radicals generated from α-oxygen carboxylic acids efficiently engage in the dearomative process of electron-deficient indoles as well (**3af-3ai**). Next, non-α-heteroatom alkyl radicals bearing four-, five- and six-rings were probed. The corresponding products **3aj**, **3ak**, **3am**, **3ao** and **3aq** were delivered in high yields. This protocol was also successfully expanded to bridged carboxylic acids (**3an** and **3ap**) as alkyl radical precursors. As for hindered structures and less reactive radicals, stronger light power and a more amount of acid are needed to achieve good yields (**3ad**, **3ae**, **3aj–3aq**). It should also be pointed out that under the mild reaction conditions, this radical-based method exhibits broad functional group tolerance, as demonstrated for free hydroxyl (**3y**), acetal (**3ai**), thioether (**3ab**), amide (**3aa**), and heteroaromatic groups (**3x** and **3z**). Especially, electron-deficient indole was chemoselectively reacted in the presence of electron-rich indole, demonstrated by the case of **3x**. Furthermore, a dipeptide can also effectively participate in

**Table 1 Optimization of reaction conditions.**

| Entry | PC (5 mol%) | Solvent (0.1 M) | Time (h) | Yield (%)[a] |
|---|---|---|---|---|
| 1 | (Ir[dF(CF₃)ppy]₂(dtbpy))PF₆ | DMF | 36 | 56 |
| 2 | 4CzIPN | DMF | 36 | 99[b] |
| 3 | 4CzIPN | MeCN | 36 | 81 |
| 4 | 4CzIPN | DMF | 24 | 75 |
| 5 | 4CzIPN | DMF | 24 | 83[c] |
| 6[d] | 4CzIPN | DMF | 36 | 91[d] |
| 7 | None | DMF | 36 | NR[e] |
| 8 | 4CzIPN | DMF | 36 | NR[f] |

Unless otherwise specified, to an oven-dried 10 mL-Schlenk tube equipped with a stir bar, was added 1a (55.0 mg, 0.2 mmol), 2a (55.0 mg, 0.2 mmol), PC (8.0 mg, 5 mol%), 1b (45.5 mg, 0.26 mmol), Cs₂CO₃ (65.0 mg, 0.2 mmol) and solvent (2.0 mL). The mixture was degassed by freeze-pump-thaw method, then sealed with parafilm. The solution was then stirred at rt under the irradiation of a 5 W blue LED strip for the indicated time.
PC photoredox catalyst, DMF dimethylformamide, NMP N-Methyl-2-pyrrolidone.
[a]¹H NMR yield.
[b]Isolated yield.
[c]0.1 mmol Cs₂CO₃ was used.
[d]3 mol% PC was used.
[e]No reaction.
[f]No light.

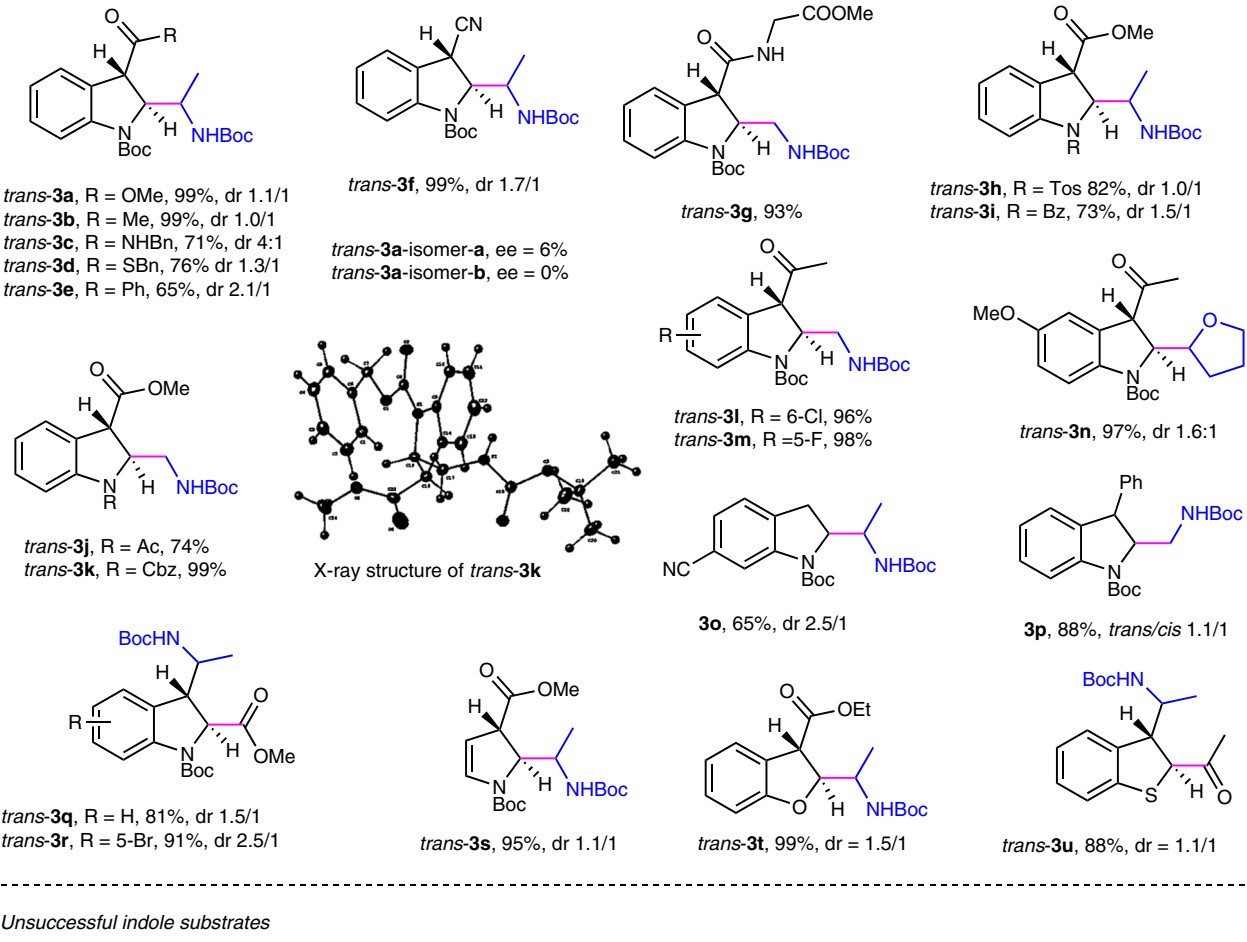

**Fig. 2 Scope of indoles and other hetereoaromatics.** Unless specified, see the general procedure in Supplementary Methods Section 1.5 for the experimental protocol.

this process. In the scope study, we found that four unsuccessful carboxylic acids could not partipatite in the process (Fig. 3).

**Late stage dearomatization of indoles and gram-scale synthesis of indolines**. To further demonstrate the utility of this mild dearomatization strategy, we performed a series of late-stage modifications on natural products. As shown in Fig. 4a (Supplementary Methods Section 1.5), the standard protocol was successfully applied to natively and selectively modify natural products (+)-menthol, cholesterol and (+)-δ-tocopherol to give *trans*-indoline-based analogues **3as**, **3aw** and **3ax** in 99, 61 and 80% yield, respectively. Moreover, pentose and hexose derived indoles gave the desired products **3at − 3au**. Finally, *trans*-indoline containing dipeptide **3av** was prepared in high yield (83%). This approach can be applied in a gram scale (2 mmol) synthesis of indolines without loss of yields (Figs. 4b, 3k and 3aj)

(Supplementary Methods Section 1.5). Moreover, the obtained indolines can go further transformation such as deprotection of *N*-Boc indoline and reduction of methyl ester to alcohol (**4k** and **4aj**) (Supplementary Methods Section 1.5 and Supplementary Data 1).

**Asymmetric dearomatization of indoles**. With the success of efficient radical engaged dearomatization of indoles and other heteroaromatics, we seek to realize the asymmetric version of this synthetically useful approach. The development of radical engaged reactions including asymmetric dearomatization of indoles has been a formidably challenging task in photoredox catalysis and is much less developed, but highly sought area[58,59]. It is observed that so far, all reported but limited asymmetric examples employ the electron-rich indoles[18,22,60–62]. In this study, we proposed to realize the asymmetric manner using a

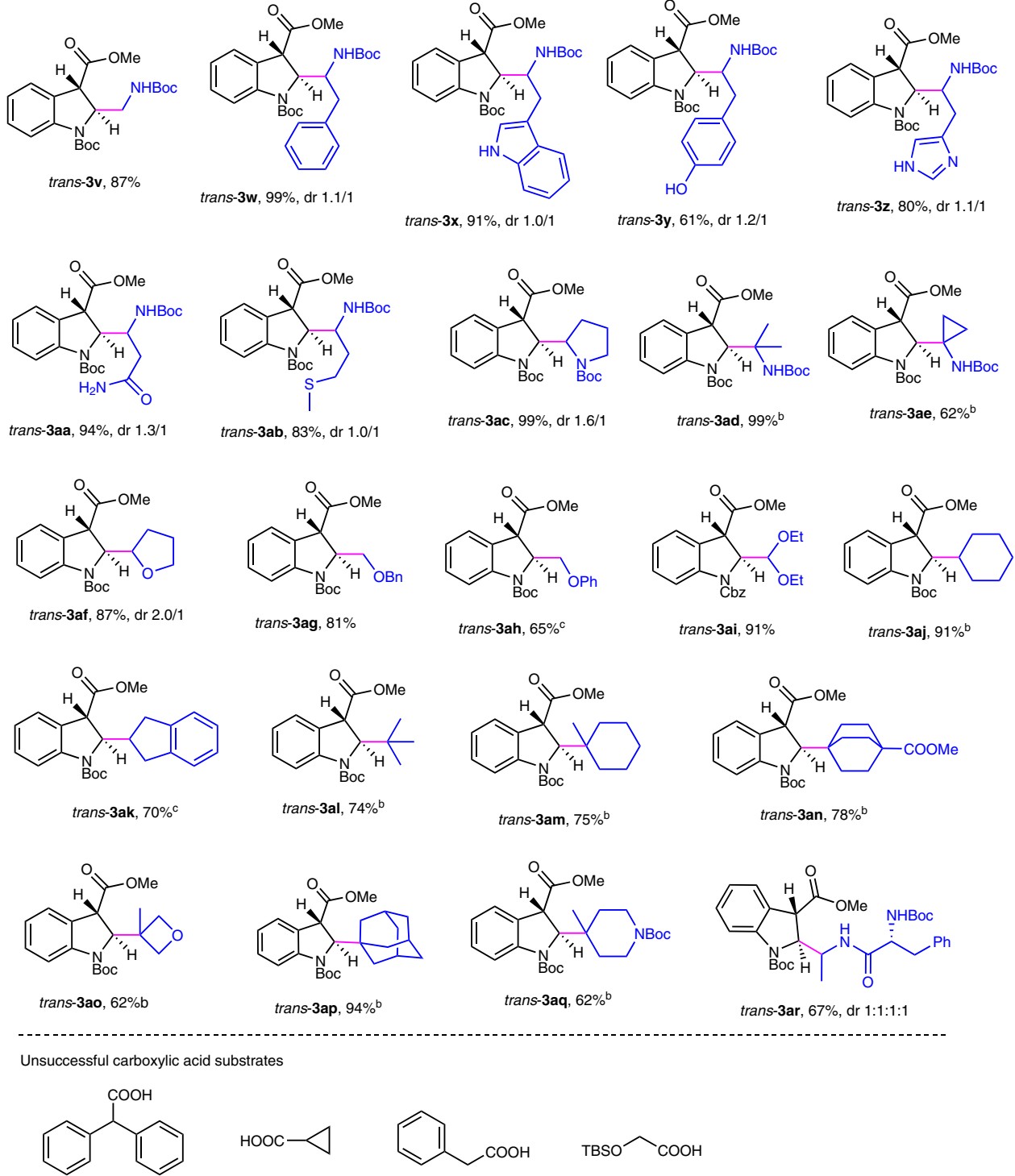

**Fig. 3 Scope of carboxylic acids.** Unless specified, see the general procedure in Supplementary Methods Section 1.5 for the experimental protocol.

[a]For all cases, product is a racemic mixture. Yields refer isolated yields after column chromatography. dr was determined by [1]H NMR of crude mixtures. [b]carboxylic acid (0.40 mmol, 2.0 equiv) and two 40W Kessil blue LED lamps were used. [c]carboxylic acid (0.30 mmol, 2.0 equiv) and two 40W Kessil blue LED lamps were used.

chiral auxiliary induced chirality strategy because the carboxylate can be served as a handle for the incorporation of a chiral auxiliary[63]. However, they are rarely employed in radical processes[64,65] probably because of the high reactivity of radial species.

Commonly used Evans chiral oxazolidinone auxiliary[66] was explored in our initial attempt. Under the above established

optimal reaction conditions, the reaction of chiral oxazolidinone (**5a**) with *N*-Boc alanine (**2b**) gave the product in quantitative yield but only poor distereoselectivity (1.6:1 dr, Table 2, entry 1). It appears that solvent has little effect on the distereoselectivity (entries 1–3), but noticed impact on reaction yield (entries 1, 2 and 5). As Lewis acid (LA) is often used to improve dr by reducing the rotation of the auxiliary through chelating two C = O bonds,

**a. Late stage dearomatization of indoles**

(+)-menthol derivative
*trans*-**3as**, 99%, dr 1.0/1

ribose derivative
*trans*-**3at**, 70%, dr 1.0/1

galactose derivative
*trans*-**3au**, 81%, dr 1.0/1

Phe-Glu derivative
*trans*-**3av**, 83%, dr 1.1/1

Cholesterol derivative
*trans*-**3aw**, 61%, dr 1.0:1

(+)-δ-Tocopherol derivative
*trans*-**3ax**, 80%, dr 1.0/1

**b. Large scale synthesis and synthetic elaboration**

For all cases, product is a racemic mixture; yields refer isolated yields after column chromatography; dr was determined by $^1$H NMR of crude mixture.

**Fig. 4 Late-stage dearomatization of indoles and gram-scale synthesis of indolines. a** Late-stage dearomatization of indoles. **b** Large scale synthesis and synthetic elaboration.

we screened several LAs. However, no enhancement of dr value was observed, which probably attributes to the disruption of the chelation interaction by highly polar solvent DMF (entries 6–9). Switching to *L*-proline derived indole **5b** did not produce an encouraging result (entry 10). Gladly, when chiral camphorsultam[67] was employed as a chiral auxiliary, excellent dr value (>20:1) was obtained in high yield (99%, entry 11). Shortening reaction time decreased the yield slightly (entry 12). These studies led to a protocol for the asymmetric dearomatization of electron-deficient indoles for the first time and provide an efficient approach to the synthesis of medicinally valued chiral indoline derived amino acids (β- and/or γ-amino acids).

Next, we evaluated the scope of the asymmetric process under the optimized reaction conditions (Fig. 5) (Supplementary Methods Section 1.6 and Supplementary Data 1 and 2). To get more accurate dr value of dearomatization indoline products, the camphorsultam auxiliary was removed by either hydrolysis (Condition A) or reduction (Condition B) to analyze the ee value of derived compounds such as ester or alcohol using chiral HPLC. The results reveal that this radical engaged asymmetric strategy serves as a general approach to enantioenriched *trans* 2,3-disubstituted indolines **7** with high enantioselectivity (up to 98% ee). α-Amino alkyl groups were successfully introduced to deliver pharmaceutically valued chiral β,γ-diamino acids with

**Table 2 Optimization of asymmetric dearomatization reaction conditions.**

| Entry | Substrate | Additive | Solvent (0.1 M) | Yield(%)[a] | Dr[b] |
|---|---|---|---|---|---|
| 1 | **5a** | None | DMF | 99[c] | 1.6:1 |
| 2 | **5a** | None | MeCN | 78 | 1.3:1 |
| 3 | **5a** | None | NMP | 99 | 1.6:1 |
| 4[d] | **5a** | None | DMF | 99 | 1.8:1 |
| 5[d] | **5a** | None | EtOAc | <5 | Not determined |
| 6 | **5a** | BF$_3$ (10 mol%) | DMF | 61 | 1.4:1 |
| 7[e] | **5a** | Zn(OAc)$_2$ (1.0 equiv.) | DMF | <5 | Not determined |
| 8[e] | **5a** | Mg(OAc)$_2$ (1.0 equiv.) | DMF | 52 | 1.5:1 |
| 9[e] | **5a** | LiOAc (1.0 equiv.) | DMF | 65 | 1.4:1 |
| 10 | **5b** | None | DMF | 77 | 1.1:1 |
| 11 | **5c** | None | DMF | 99[c] | >20:1 |
| 12 | **5c** | None | DMF | 91%[f] | >20:1 |

Unless otherwise specified, to an oven-dried 10 mL-Schlenk tube equipped with a stir bar, was added **5** (0.1 mmol), photoredox catalyst (4CzIPN, 4.0 mg, 5 mol%), **2b** (23.0 mg, 0.13 mmol), Cs$_2$CO$_3$ (32.0 mg, 0.1 mmol) and solvent (1.0 mL). The mixture was degassed by freeze-pump-thaw method, then sealed with parafilm. The solution was then stirred at rt under the irradiation of a 5 W blue LED strip for 36 h.
*DMF* dimethylformamide, *NMP* N-Methyl-2-pyrrolidone.
[a]$^1$H NMR yield.
[b]Determined by $^1$H NMR.
[c]Isolated yield.
[d]CsOAc (1.0 equiv.) used as base.
[e]No base used.
[f]Reaction time: 24 h.

**Fig. 5 Scope of asymmetric dearomatization of indoles.** Unless specified, see the general procedure in Supplementary Methods Section 1.6 for the experimental protocol.

both high ee values and yields by using natural amino acids (**7a–d**). Besides, radicals produced from α-oxygen carboxylic acids reacted with indole substrates smoothly (**7e–m**). Notable, various functional groups were tolerated well under the reaction

conditions such as allylic (**7e**), propargyl (**7f**), benzyl (**7h**) and halogens (**7j** and **7k**). Moreover, indolines bearing heteroaromatics were successfully obtained (**7m–o**). Again, the radical approach enables the incorporation of highly sterically

demanding structures (**7p–v**), which are particularly challenging using catalytic ionic methods with high level of enantioselectivity. We also probed indoles with different protecting groups (PGs) on nitrogen. It was found that Boc and Cbz (**7w**) were untouched while other PGs such as Ac, Bz, Tos, and pivaloyl were sensitive to the reaction conditions. It seems that chiral moieties do not affect the newly formed stereogenic centers, as seen in the synthesis of chiral indoline saccharide derivatives with both good ee value and yields (**7x** and **7y**). The gram scale (1 mmol) synthesis of **8c** was also realized in 72% yield and 94% ee and the camphorsultam auxiliary was recovered at the same time (81% yield) (Supplementary Methods Section 1.7). The absolute conformation of the products **7** and **8** were determined to 2*R* and 3*S* by converting a known chiral compound (Supplementary Methods Section 1.7)[68]. In all cases except products **8a–c** as alcohols, chiral methyl esters **7** were used for chiral HPLC analysis. Finally, unsuccessful substrates provided in Fig. 5 reveal the limitations of this methodology.

## Discussion

A Giese-type process has been successfully implemented for the dearomatization of electron-poor indole systems in this study. The method can also be viewed as a decarboxylative Michael addition process, which has been intensively studied in recent years[35,50–55]. However, indoles are not the same as simple alkenes and to the best of our knowledge, breaking the stable $C_2=C_3$ bond embedded in the aromatic indoles with this strategy has not been reported. Furthermore, in a typical decarboxylative Michael addition process[35,50–55], less hindered electron-deficient vinyls are generally used for effective transformation, whereas this process accomplishes with more complicated α, β-unsaturated systems. Therefore, the process significantly expands the scope of the synthetic strategy. Moreover, the process offers a distinct approach to highly vaued indolines. The dearomative structures are complementary to those of well studied electron-rich systems. The tethered EWGs such as ester, ketone, amide, nitrile, etc are versatile handles for further synthetic elaboration.

It is noted that although intermolecular radical addition to the indole $C_2=C_3$ double bond has been documented[48,49], these processes also use electron-rich structures. Mechanistically, our dearomatization activation strategy is completely different from that of these radical engaged Friedel-Crafts type methods[48,49]. In our approach, an oxidative process is implemented to generate a nucleophilic radical for an addition to an electron-deficient C=C bond (Fig. 6a). The process delivers a dearomative product. In contrast, an opposite photoredoxcatalytic reductive activation produces an electrophilic radical for reacting with the electron-rich indole $C_2=C_3$ double bond. Therefore, an aromatic product is obtained instead via subsequent oxidation to accomplish the catalytic cycle[48,49].

What we learned from this study is that the successful realization of the distinct indole dearomatization method lies in the rationalization of the reactivities of the radical species in organophotoredox catalytic cycle and enables achieving the chemoselectivity. In the intensively studied photoredox mediated indole dearomatization processes, direct oxidation of electron-rich indoles can be achieved because they have relatively low redox potentials. For example, $E_{redox}$ of *N*,3-dimethyl indole is ca. +0.4 V vs SCE[34]. However, indoles bearing EWGs at N, $C_2$ or $C_3$ positions have much higher $E_{redox}$. For instance, *N*-methyl 3-acetyl indole is ca. +1.0 V vs SCE[34] while methyl *N*-Boc-3-indole carboxylate (**1a**, $E_{redox}$ + 1.94 vs SCE, see Supplementary Methods Section 1.8) is even bigger. The significant difference suggests oxidative dearomatization of electrophilic indoles is difficult. The overlooked, reversed reactivity offers a unique opportunity for

developing distinct dearomative methods, which have been successfully carried out in this study. Critically, the selective controlling reactivity makes the process possible (Fig. 6a). The excited 4CzIPN* ($E^*_{redox}$ + 1.35 vs SCE)[57] can selectively oxidize *N*-Boc-alanine Cs salt **I** (as a representative example, $E_{redox}$ + 1.00 vs SCE)[56] without crossover oxidation of methyl *N*-Boc-3-indole carboxylate (**1a**, $E_{redox}$ + 1.94 vs SCE). The resulting nucleophilic radical **II** then undergoes a Giese-type addition process with **1a** to produce radical **III**. The radical **III** is facial selectively reduced by 4CzIPN$^{•−}$ to give the *trans*- anion **IV** and 4CzIPN to complete the redox cycle. Finally, protonation of anion **IV** delivers observed *trans*-dearomatization product **3a**. The pathway is consistent with the photoredox Giese reaction. Furthermore, our control experiments support the proposed pathway (Supplementary Methods Section 1.9). The radical engaged process is verified by a radical scavenger TEMPO suppressed reaction (Fig. 6b). The anion intermediate **IV** is validated by a $D_2O$ quenching experiment. To get more insights of the reactivity of **1a** compared with the classic Michael acceptor in Giese reaction, a competition experiment using **1a** and cyclopent-2-en-1-one reacting with *N*-Boc-glycine was performed. Interestingly, we didn't observe *trans*-**3v**, but only product **12** coming from cyclopent-2-en-1-one with a yield of 87%. This result shows the much less reactivity of **1a** than commonly used electron-deficient olefins.

Besides, we were interested in how the chiral auxiliary induces diasteroselectivity. Based on Curran's work[69,70], a model for the asymmetric dearomatization was proposed (Fig. 6c). The sulfonyl group, which is spatially located close to C2, is believed to play a major role in the inducement of asymmetric selectivity. Radical attacking C2 from the *Re* face(bottom) is favored due to less steric interaction between radical and equatorial β oxygen of the sulfonyl group. While axial α oxygen blocks the radical addition from the *Si* (top) face because of their strong steric interaction. The resulting configuration of the product from this model is consistent with our experimental results.

## Conclusion

In summary, we have developed an unprecedented versatile organophotoredox process for the dearomatization of electron-poor indoles. A distinct strategy that a photoredox mediated Giese-type transformation is introduced to break the electron-deficient aromatic C=C bonds for the first time. The preparative power of the dearomatization strategy has been demonstrated by the use of naturally abundant carboxylic acids and readily available structurally diverse indoles bearing common EWGs including (thio)ester, amide, ketone, nitrile, and even aromatics at either $C_2$ or $C_3$ positions. A wide array of 2,3-disubstituted indolines with high anti-stereoselectivity (>20:1 dr) are prepared. This powerful manifold can also be applied to other aromatic heterocycles such as pyrroles, benzofurans, and benzothiophenes for dearomatization. Furthermore, enantioselective dearomatization of indoles has been achieved by a chiral camphorsultam auxiliary with high enantioselectivity (up to 98% ee). The simplicity, efficiency, and board scope of this distinct synthetic strategy will be appreciated by organic and medicinal chemists to rapid access a library of synthetically and biologically important indolines.

## Methods

**Typical procedure for the synthesis of racemic products 3 (Figs. 2–4)**. To an oven-dried 10 mL-Schlenk tube equipped with a stir bar, was added indole derivatives (0.2 mmol, 1.0 equiv.), 4CzIPN (8.0 mg, 5 mol%), acid (0.26 mmol, 1.3 equiv.), $Cs_2CO_3$ (65.0 mg, 0.2 mmol, 1.0 equiv.) and DMF (2.0 mL). The mixture was degassed by freeze-pump-thaw method, then sealed with parafilm. The solution was then stirred at room temperature under the irradiation of a 5w blue LED strip or two 40 w blue LED lamps for 36 h. After completion of the reaction, the mixture was diluted with 20 mL of water and extracted by EtOAc (3 × 10 mL).

**a. Proposed catalytic cycle**

**c. Proposed model for asymmetric dearomatization**

**b. Reactions designed for mechanistic studies**

**Fig. 6 Proposed reaction mechanism and mechanistic studies.** See Supplementary Methods Section 1.9 for the experimental protocol.

The organic layer was collected, dried by $Na_2SO_4$ and concentrated under vacuum. The residue was purified by flash column chromatography to afford the product.

**Typical procedure for the synthesis of products 7 and 8 (Fig. 5).** To an oven-dried 10 mL-Schlenk tube equipped with a stir bar, was added chiral auxiliary

attached indole derivatives 5 (0.1 mmol, 1.0 equiv.), photocatalyst 4CzIPN (4.0 mg, 5 mol%), acid (0.13–0.2 mmol, 1.3–2.0 equiv.), $Cs_2CO_3$ (32.0 mg, 0.1 mmol, 1.0 equiv.) and DMF (1.0 mL). The mixture was degassed by freeze-pump-thaw method, then sealed with parafilm. The solution was then stirred at room temperature under the irradiation of a 5w blue LED strip or two 40 w blue LED lamps for the indicated time. After completion of the reaction, the mixture was diluted with 10 mL of water and extracted by EtOAc (3 × 5 mL). The combined organic

layers were washed by 10 mL of brine, dried by $Na_2SO_4$, and concentrated under vacuum. The residue was used in the next step without purification.

*Condition A.* the residue was dissolved in 0.8 mL of THF and 0.2 mL of water. To the solution was added LiOH (12 mg, 0.5 mmol) and $H_2O_2$ (113 μL, 30% (w/w) in water) at rt. The reaction was stirred at rt for 10 h, after which 10 mL of EtOAc was added. The mixture was washed by 0.5 M NaOH ($3 \times 5$ mL). The combined aqueous layers were collected, washed once with 5 mL of $Et_2O$ and acidified with 2 M HCl to pH = 2–3. The aqueous solution was extracted by EtOAc ($3 \times 5$ mL). The combined organic layers were dried by $Na_2SO_4$ and concentrated under a vacuum. The residue was used in next step without purification. The residue was dissolved in 0.75 mL of $Et_2O$ and 0.25 mL of MeOH. To the solution was added (trimethylsilyl)diazomethane solution (2 M, 110 μL) at 0 °C under nitrogen atmosphere, the mixture was stirred at room temperature for 20 min. The reaction mixture was concentrated in vacuo and purified by flash column chromatography or preparative TLC plate to afford the product.

*Condition B.* The residue was dissolved in 0.75 mL of EtOH and 0.25 mL of $Et_2O$. To the solution was added LiCl (21 mg, 0.5 mmol) and $NaBH_4$ (31.5 mg, 0.5 mmol) at rt. The reaction was stirred at rt for 3 h, after which the solution was concentrated and purified by column chromatography or preparative TLC plate to afford the product.

## Data availability

[1]H and [13]C NMR spectra for products 3, 7, and 8: see Supplementary Figs. 3–192 in Supplmentary Data 1.

Chiral HPLC analysis for products 7, 8, and 11: see Supplementary Figs. 193–254 in Supplmentary Data 2.

The authors declare that all the other data including Supplementary Methods and compound structural characterization data, supporting the findings of this study are available within this paper, its Supplementary Information file. The X-ray crystallographic coordinates (Supplementary Methods: Supplementary Tables 1–3 and Supplementary Table 3: CIF file) for structures *trans*-3k reported in this study have been deposited at the Cambridge Crystallographic Data Centre (CCDC), under deposition numbers CCDC-1994584. These data can be obtained free of charge from The Cambridge Crystallographic Data Centre.

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

## Acknowledgements

Financial support of this research from the University of Arizona is gratefully acknowledged. The Foundation for Basic and Applied Research of Guangdong Province (2019A1515110489) for Dr. Ziyuan Zhou; This work supported by the high-performance computing platform of Peking University.

## Author contributions

Y.Z., P.J., Y.D., H.H., C.W., Z.Z. and F.G. planned, conducted, and analyzed the experiments. W.W. planned, designed, and directed the project, and Y.Z. and W.W. wrote the manuscript.

## Competing interests

The authors declare no competing interests.
