## [Peer Review File · Communications Chemistry]

This manuscript has been previously reviewed at another Nature Research journal. This document only contains reviewer comments and rebuttal letters for versions considered at Communications Chemistry.

REVIEWERS' COMMENTS:

Reviewer #1 (Remarks to the Author):

Overall, the authors have done an admirable job of addressing the comments and suggestions by the reviewers. However, it would appear that a lot of valuable information from the additional experiments, primarily those focused on the method's limitations, were added to the SI rather than the manuscript (responses to reviewer 1 in #'s 1, 2, and 4). It is incredibly valuable to the community to know advantages and limitations. Rather than burying this information in a large SI file, this should be included in the manuscript. This can be achieved through either additional text or endnote citations that provide a reference for the reader as to what was examined and what failed. Aside from this relatively minor correction, the current submission appears suitable for publication in Nat. Commun.

January 20, 2021

Reviewer 1

1. “However, it would appear that a lot of valuable information from the additional experiments, primarily those focused on the method’s limitations, were added to the SI rather than the manuscript (responses to reviewer 1 in #'s 1, 2, and 4). It is incredibly valuable to the community to know advantages and limitations. Rather than burying this information in a large SI file, this should be included in the manuscript. This can be achieved through either additional text or endnote citations that provide a reference for the reader as to what was examined and what failed..”

Response: As suggested, we have added discussion and the unsuccessful substrates in the main manuscript to reveal the limitations of this strategy.